# 2013–2019 increases of surface ozone pollution in China: anthropogenic and meteorological influences

Ke Li[1], Daniel J. Jacob[1], Lu Shen[1], Xiao Lu[1], Isabelle De Smedt[2], and Hong Liao[3,4]

[1]John A. Paulson School of Engineering and Applied Sciences, Harvard University, Cambridge, MA, USA
[2]Belgian Institute for Space Aeronomy (BIRA-IASB), Brussels, Belgium
[3]Jiangsu Key Laboratory of Atmospheric Environment Monitoring and Pollution Control, Collaborative Innovation Center of Atmospheric Environment and Equipment Technology, School of Environmental Science and Engineering, Nanjing University of Information Science and Technology, Nanjing, China
[4]Harvard-NUIST Joint Laboratory for Air Quality and Climate, Nanjing University of Information Science and Technology, Nanjing, China

*Correspondence to*: Ke Li (keli@seas.harvard.edu)

**Abstract.** Surface ozone data from the Chinese Ministry of Ecology and Environment (MEE) network show sustained increases across the country over the 2013–2019 period. Despite Phase 2 of Clean Air Action targeting ozone pollution, ozone was higher in 2018–2019 than in previous years. The mean summer 2013–2019 trend of maximum 8-h average (MDA8) ozone was 1.9 ppb a$^{-1}$ ($p<0.01$) across China and 3.3 ppb a$^{-1}$ ($p<0.01$) in the North China Plain (NCP). Fitting ozone to meteorological variables with a multiple linear regression model shows that meteorology played a significant but not dominant role in the 2013–2019 ozone trend, contributing 0.70 ppb a$^{-1}$ ($p<0.01$) across China and 1.4 ppb a$^{-1}$ ($p=0.02$) in the NCP. Rising June-July temperatures over the NCP were the main meteorological driver, particularly in recent years (2017–2019), and were associated with increased foehn winds. NCP data for 2017–2019 show a 15% decrease in fine particulate matter (PM$_{2.5}$) that may be driving the continued anthropogenic increase in ozone, and unmitigated emissions of volatile organic compounds (VOCs). VOC emission reductions, as targeted by Phase 2 of the Chinese Clean Air Action, are needed to reverse the increase of ozone.

# 1 Introduction

Surface ozone is a serious air pollution issue over much of eastern China (Ma et al., 2012; Fu et al., 2019). Measurements from the Chinese Ministry of Environment and Ecology (MEE) network of sites frequently exceed the national air quality standard of 160 μg m$^{-3}$, corresponding to 82 ppb at 298 K and 1013 hPa (Li et al., 2017; Shen et al., 2019a; Fan et al., 2020). The Clean Air Action initiated in 2013 imposed rapid decreases in pollutant emissions (Chinese State Council, 2013) and resulted in large decreases in fine particulate matter (PM$_{2.5}$) concentrations (Zhai et al., 2019; Q. Zhang et al., 2019). However, ozone increased by 1–3 ppb a$^{-1}$ over the 2013–2017 period in megacity clusters of eastern China (Lu et al., 2018; Li et al., 2019a; Lu et al. 2020), partly offsetting the health benefits from improved PM$_{2.5}$ (Dang and Liao, 2019; Q. Zhang et al., 2019). Phase 2 of Clean Air Action starting in 2018 (Chinese State Council, 2018) imposed new emission controls targeted at ozone. Here we show that the increasing ozone trend in eastern China has continued through 2019, driven by both anthropogenic emission and meteorological trends, and stressing the urgent need for more vigorous emission controls.

Ozone in polluted regions is produced by photochemical reactions of volatile organic compounds (VOCs) and nitrogen oxides (NO$_x \equiv$ NO + NO$_2$), enabled by hydrogen oxide radicals (HO$_x \equiv$ OH + peroxy radicals) as oxidants. VOCs and NO$_x$ are emitted by fuel combustion, and VOCs have additional industrial sources (Zheng et al., 2018) and biogenic sources (Guenther et al., 2012). HO$_x$ is produced photochemically from ozone and water, formaldehyde (HCHO), nitrous acid, and other precursors (Tan et al., 2019). Ozone is highest in summer when photochemistry is most active (Wang et al., 2017). Meteorological conditions play an important role in modulating ozone concentrations, not only through transport but also by affecting natural emissions and chemical rates (Jacob and Winner, 2009; Shen et al., 2016; Fu et al., 2019; Lu et al., 2019).

A number of studies have investigated the roles of anthropogenic and meteorological factors in driving the 2013–2017 ozone trend, and concluded that meteorological factors were not negligible but anthropogenic factors were dominant (Ding et al., 2019; Li et al., 2019a; Liu et al., 2019; Yu et al., 2019; Liu et al., 2020). Our previous work (Li et al., 2019a, 2019b) found that the decrease of PM$_{2.5}$ was a major factor driving the increase of ozone due to the role of PM$_{2.5}$ as scavenger of hydroperoxy (HO$_2$) radicals and NO$_x$ that would otherwise produce ozone. Here we extend the analysis of ozone trends to 2019, into the implementation of Clean Air Action Phase 2, and bring in satellite and ground-based observations to relate the most recent ozone trends to those of VOC (Shen et al., 2019b) and NO$_x$ emissions (Zheng et al., 2018; Shah et al., 2020).

## 2 Data and methods

### 2.1 Surface measurements

Hourly concentrations of ozone, PM$_{2.5}$, and NO$_2$ are taken from the MEE website (http://106.37.208.233:20035) and archived at http://beijingair.sinaapp.com. The network was launched in 2013 as part of the Clean Air Action. It included 450 monitoring stations in 2013, growing to ~1500 stations by 2019. In this study, ozone trends are estimated across all the sites including those with partial records. We will show later that the estimated ozone trends change only marginally if continuous records throughout 2013–2019 are used in the analysis. We compute maximum daily 8-h average (MDA8) ozone as well as 24-h average PM$_{2.5}$ and NO$_2$ concentrations from the hourly data for June-July-August (JJA). Concentrations were reported by the MEE in units of µg m$^{-3}$ under standard conditions (273 K, 1013 hPa) until 31 August 2018. This reference state was changed on 1 September 2018 to (298 K, 1013 hPa) for gases and local ambient state for PM$_{2.5}$ (MEE, 2018). We converted ozone and NO$_2$ concentrations to ppb, and rescaled post-August 2018 PM$_{2.5}$ concentrations to standard conditions by assuming (298 K, 1013 hPa) as the local ambient state.

### 2.2 Satellite observations

We use observations of NO$_2$ and formaldehyde (HCHO) columns from the OMI and TROPOMI satellite instruments to track recent changes in anthropogenic emissions of NO$_x$ and VOCs, respectively. Shen et al. (2019b) and Shah et al. (2020) previously found that OMI-derived trends of VOC and NO$_x$ emissions were consistent with 2013–2017 bottom-up estimates from the Multi-resolution Emission Inventory for China (MEIC; Zheng et al., 2018). Here we extend the analysis using 2013–2019 OMI data from the European Quality Assurance for Essential Climate Variables project for NO$_2$ (Boersma et al., 2018) and HCHO (De Smedt et al., 2015). We further use TROPOMI HCHO data available for the summers of 2018–2019 (De Smedt et al., 2018). We do not use TROPOMI NO$_2$ data because of a version change in March 2019 from v1.2.0 to v1.3.0 that could bias the trend between the summers of 2018 and 2019 (https://sentinel.esa.int/documents/247904/2474726/Sentinel-5P-Level-2-Product-UserManual-Nitrogen-Dioxide, last access: 20 July 2020). The TROPOMI HCHO data are freely accessed from https://s5phub.copernicus.eu/dhus/ (last access: 28 February 2020) and we only use observations with quality assurance value larger than 0.5. This filter effectively removes data with cloud fraction larger than 0.5. Interannual trends in HCHO columns could be affected by temperature-dependent emissions of biogenic VOCs (Palmer et al., 2006). Following Zhu et al. (2017), we remove this contribution by regressing JJA monthly mean HCHO columns onto noon (13:00 local time) surface air temperatures, and then subtracting this fitted temperature dependency.

## 2.3 Stepwise multiple linear regression (MLR) model

To quantify the role of meteorology in driving 2013–2019 ozone trends, we use the same stepwise multiple linear regression (MLR) modeling approach as Li et al. (2019a). This modeling approach relates the month-to-month variability of MDA8 ozone to that of meteorological variables. Consistent meteorological fields for 2013–2019 were obtained from
the NASA Modern-Era Retrospective Analysis for Research and Applications, Version 2 (MERRA-2) product (Gelaro et al., 2017). The MERRA-2 data have a spatial resolution of 0.5° latitude × 0.625° longitude. We average the daily MDA8 ozone from the MEE network onto the MERRA-2 grid. Firstly, the regression model is applied to select the key meteorological parameters driving the day-to-day variability of ozone for each grid cell. There are nine MERRA-2 meteorological variables considered as ozone covariates, including daily maximum 2-m air temperature (Tmax), 10-m
zonal wind (U10) and meridional wind (V10), boundary layer height (PBLH), total cloud area fraction (TCC), rainfall (Rain), sea level pressure (SLP), relative humidity (RH), and 850-hPa meridional wind (V850), following (Li et al., 2019a). The meteorology fields are averaged over 24 h for use in the MLR model except for PBLH and TCC, which are averaged over daytime hours (8–20 local time), and for Tmax (daily maximum).

Secondly, to avoid overfitting, only the three locally dominant meteorological parameters are regressed onto the
deseasonalized monthly MDA8 ozone to fit the role of 2013–2019 meteorological variability. The top three variables are selected based on their individual contribution to the regressed ozone, along with the requirement that they are statistically significant above the 95% confidence level in the MLR model. They will differ for each 0.5° × 0.625° grid cell. We show these top three meteorological drivers for ozone variability in Figure S1–S3 for different locations in China.

Thirdly, we fit the observed monthly ozone anomalies by applying these dominant meteorological drivers in the MLR model. The coefficients of determination ($R^2$) for the MLR model are generally above 0.4–0.5 for polluted regions of China which are of most interest to us (Figure S4). Remote locations with background ozone levels have less ozone variability and are thus harder to fit. Similar MLR models have been extensively employed to quantify the effect of meteorological variability on air pollutants (e.g., Tai et al., 2010; Otero et al., 2018; Zhai et al., 2019; Han et al., 2020).

Finally, the trend in regressed ozone is taken to reflect the meteorological contribution, and the residual is then taken to reflect the presumed anthropogenic contribution, with the statistical significance of the anthropogenic trend determined by Student's $t$-test. We have followed this approach before to isolate the anthropogenic trends of ozone and $PM_{2.5}$ (Li et al., 2019a; Zhai et al., 2019). Similar statistical decomposition of anthropogenic and meteorological contributions to air pollutant trends has been employed by previous studies (e.g., Chen et al. 2019; Yu et al., 2019; X. Zhang et al., 2019).
The effect of biogenic VOCs on ozone trends depends on meteorological and land cover drivers. Meteorological drivers,

in particular temperature, would be accounted for in the MLR model. The effect of land cover changes is expected to be small over the 7-year time horizon of our analysis (Fu and Tai, 2014)

## 3 Results and discussion

We first present the general 2013–2019 summer ozone trends in China and their statistically decomposed meteorological and anthropogenic contributions. Ozone trends over the major megacity clusters in China are highlighted. We go on to more specifically attribute the meteorological and anthropogenic drivers of recent ozone trends over the North China Plain, where the ozone increase is the highest.

### 3.1 2013–2019 ozone trends: anthropogenic and meteorological contributions

**Figure 1** shows 2013–2019 trends of summer maximum and mean MDA8 ozone and $PM_{2.5}$ from the MEE network. The Clean Air Action has dramatically improved $PM_{2.5}$ pollution since 2013, with ~50% decrease of summertime mean $PM_{2.5}$ concentrations across eastern China over the 2013–2019 period. Maximum $PM_{2.5}$ concentrations experienced a similar decrease trend. In contrast, ozone has been steadily increasing over 2013–2019 and concentrations in 2019 are the highest in the record. The Clean Air Action focused specific attention on the four megacity clusters identified by rectangles: North China Plain (NCP, 34°–41°N, 113°–119°E), Yangtze River Delta (YRD, 30°–33°N, 119°–122°E), Pearl River Delta (PRD, 21.5°–24°N, 112°–115.5°E), and Sichuan Basin (SCB, 28.5°–31.5°N, 103.5°–107°E). Mean MDA8 ozone in summer 2019 averaged 83 ppb across the NCP sites and maximum MDA8 ozone averaged 129 ppb. Summer mean MDA8 ozone in 2019 was lower for the other megacity clusters (67 ppb for YRD, 46 ppb for PRD, and 57 ppb for SCB) but summer maximum MDA8 ozone values were comparable to the NCP. These three megacity clusters are subject to similar ozone pollution episodes under stagnant conditions as the NCP (Wang et al., 2017), but they are more frequently ventilated by the summer monsoon bringing cleaner tropical air and precipitation hence the lower mean ozone.

**Figure 2a** shows the 2013–2019 trends in summer mean MDA8 ozone obtained by ordinary linear regression of the data averaged over the 0.5° × 0.625° MERRA-2 grid. Ozone increases almost everywhere in China. Decreases are largely restricted to the Shandong Peninsula and northeastern China (including Heilongjiang, Jilin, and Liaoning provinces). The mean trend for China is 1.9 ppb $a^{-1}$ (p<0.01). Trends in the four megacity clusters are 3.3 ppb $a^{-1}$ (p<0.01) for NCP, 1.6 ppb $a^{-1}$ (p<0.01) for YRD, 1.1 ppb $a^{-1}$ (p=0.03) for PRD, and 0.7 ppb $a^{-1}$ (p=0.23) for SCB (Table 1). The increases are largest in the NCP, which could be explained by greater influence of radical scavenging by $PM_{2.5}$ (Li et al., 2019a, 2019b).

**Figure 2b** shows the meteorologically driven ozone trends, as determined by fitting ozone to meteorological variables with the MLR model. We find an average meteorologically driven trend of 0.7 ppb a$^{-1}$ (p<0.01) for China. Ozone trends over 2013–2019 in the NCP and PRD are significantly contributed by meteorology, and this is particularly driven by 2018–2019 (Table 1). Similar to our previous study for 2013–2017 (Li et al., 2019a), the most important meteorological
predictor variables in the MLR model are daily maximum temperature for the NCP and meridional wind at 850 hPa for the PRD (Figure S1). These dominant meteorological parameters are also consistent with the findings from other studies (Gong and Liao, 2019; Wang et al., 2019; Han et al., 2020). Hot weather is the main meteorological driver for high ozone in the NCP, and we will elaborate on this in the next section. The main meteorological driver for the ozone increase in the PRD is the weakening of the summer monsoonal flow (**Figure 3**) that ventilates the PRD with marine air.

On the other hand, we find that meteorology mitigated ozone pollution increases over northeastern China and the Shandong Peninsula. Summer ozone in the Shandong Peninsula is strongly affected by maritime inflow (Figure S2, J. Zhang et al, 2019; Han et al., 2020) which increased over the 2013–2019 period (**Figure 3**). Temperature decreased over northeastern China (**Figure 3**).

Removing the meteorological contribution in the ozone trend leaves a residual trend that we interpret as anthropogenic
(**Figure 2c**), following Li et al. (2019a) and Zhai et al. (2019). This anthropogenic trend is more uniformly positive at a national scale than the observed and meteorologically driven trends. It averages 1.2 ppb a$^{-1}$ (p<0.01) for all of China, as compared to 0.7 ppb a$^{-1}$ (p<0.01) for the meteorologically driven trend. The observed 2013–2019 ozone increase in all the megacity clusters except the PRD is dominated by the anthropogenic contribution, averaging 1.9 ppb a$^{-1}$ (p<0.01), 0.9 ppb a$^{-1}$ (p<0.01), and 1.0 ppb a$^{-1}$ (p<0.01) for the NCP, YRD, and SCB, respectively. This result of estimated trends still
stands if only continuous records throughout 2013–2019 are used in the analysis (Figure S5). The ozone increase in PRD is mainly meteorologically driven due to reduced monsoonal winds (**Figure 3**). The following sections present further analysis of the 2013–2019 ozone trend in the NCP, where both meteorological and anthropogenic contributions are particularly large.

**3.2 Meteorologically driven 2013–2019 ozone increase in the North China Plain**

Separating the observed 2013–2019 ozone trends by month shows that the seasonal JJA trend of 3.3 ppb a$^{-1}$ (p<0.01) over the NCP (**Figure 2a**) is driven by June and July. Observed trends are 5.5 ppb a$^{-1}$ (p<0.01) for June, 3.7 ppb a$^{-1}$ (p<0.01) for July, and 0.9 ppb a$^{-1}$ (p=0.34) for August. This month-to-month difference is mainly driven by meteorology. As derived from the MLR model, the meteorologically driven ozone trend of 1.4 ppb a$^{-1}$ (p=0.02) for JJA breaks down to 3.1 ppb a$^{-1}$ (p<0.01) for June, 2.2 ppb a$^{-1}$ (p=0.08) for July, and –1.0 ppb a$^{-1}$ (p=0.16) for August. The residual anthropogenic

trend is much more similar across months (2.4 ppb a$^{-1}$ (p=0.02) in June, 1.5 ppb a$^{-1}$ (p=0.07) in July, 1.9 ppb a$^{-1}$ (p<0.01) in August), as would be expected after removing meteorological influence.

**Figure 4** shows the monthly mean time series of daily maximum temperature averaged over the NCP for 1980–2019, with 2013–2019 highlighted in shading. Temperature is the principal driver of the meteorologically driven ozone trend as indicated by the MLR model. We find a large increase in temperature for 2013–2019 in June (0.42 °C a$^{-1}$), a lesser increase in July (0.22 °C a$^{-1}$), and a decrease in August (–0.18 °C a$^{-1}$), reflected in the meteorologically driven ozone trend for each month. When placed in the context of the 1980–2019 record, we see that the 2013–2019 temperature trends reflect interannual climate variability rather than a long-term warming trend.

Hot weather in the NCP in the summer is generally driven by large-scale anticyclonic conditions, and this has been viewed as the principal predictor of ozone pollution days (Gong and Liao, 2019). Foehn wind conditions, featuring warm and dry air subsiding from the mountains that are to the north and west of the NCP (Chen and Lu, 2016), also lead to high ozone pollution in the NCP. Foehn winds are most important in June. Following Chen and Lu (2016), we diagnosed foehn conditions in the NCP with a foehn index defined by the 850 hPa northwesterly wind averaged along a section from (42°N, 108°E) to (38°N, 112°E) (**Figure 5**). The days with positive (negative) foehn index are taken as foehn (no-foehn) condition, and one-third of summer days have positive foehn index. We find that foehn conditions are largely responsible for the 2013–2019 increase in temperature in June (**Figure 5**). The frequency of foehn conditions under hot days (>30°C) in June increased by 85% over the 2013–2019 period (driven mainly by the increased frequency in 2018–2019), and ozone increase under foehn conditions is 1.2 ppb a$^{-1}$ larger than under no-foehn conditions. Our result highlights the previously unrecognized effect of foehn winds on ozone pollution in the NCP.

### 3.3 Anthropogenically driven 2013–2019 ozone increase in the North China Plain

**Figure 6a** shows the observed time series of monthly mean JJA MDA8 ozone anomalies for 2013–2019 relative to the JJA 2013–2019 mean, averaged over all MEE sites in the NCP and including sites with partial records. We see large month-to-month variability superimposed on the long-term trend. Much of this month-to-month variability can be attributed to meteorological factors using the MLR model (blue line), as discussed in the previous section. The residual anthropogenic trend (red line) shows a 2013–2019 increasing trend with much less month-to-month variability than the original observed time series. The standard deviation decreases from 8.8 ppb to 5.3 ppb after removal of meteorological influence.

**Figure 6b** shows the 2013–2019 observed trends of different quantities relevant to the anthropogenic ozone trend over the NCP: PM$_{2.5}$ and NO$_2$ from the MEE network, and NO$_2$ and HCHO tropospheric columns from satellites. PM$_{2.5}$ shows

a steady decrease, 49% over the 2013–2019 period. $NO_2$ (a proxy for $NO_x$ emissions; Zheng et al., 2018) shows a 25–30% decrease with some interannual variability that is consistent between the OMI satellite data and the surface MEE network. HCHO (a proxy for VOC emissions) shows no significant trend for the 2013–2019 period, with some interannual variability that could reflect noise in the measurement (Shen et al., 2019b).

Of particular interest are the trends for 2017–2019, extending beyond the currently available MEIC emission inventory (Zheng et al., 2018) and during which we find continued increase of ozone. Relative to 2017, we find for 2019 a 15% decrease in $PM_{2.5}$, a 6–10% decrease in $NO_x$ emissions (depending on which proxy record we use), and flat VOC emissions.  Phase 2 of the Chinese government's Clean Air Action (China State Council, 2018) called for a 18% decrease in $PM_{2.5}$ over 2015–2020, a 15% decrease in $NO_x$ emissions, and a 10% decrease in VOC emissions. Taking into account

the already-achieved 2015–2017 gains in $PM_{2.5}$ and $NO_x$ emissions, Li et al. (2019b) inferred that those targets would require 2017–2020 decreases of 8% for $PM_{2.5}$, 9% for $NO_x$ emissions, and 10% for VOCs emissions. They found from model simulations that the decrease in $PM_{2.5}$ would cause further increase in ozone, but that decreasing VOC emissions would compensate and enable net improvement, with $NO_x$ emission changes having relatively little effect. We find here that the observed 2017–2019 decrease in $PM_{2.5}$ goes beyond the Clean Air Action target, while the satellite HCHO data

show no evidence of a decrease in VOC emissions. Combination of these two effects is consistent with the observed anthropogenically driven increase in ozone over 2017–2019. Decrease of VOC emissions is the key to reverse the ozone increase (Li et al., 2019b). Strict control measures on solvent use and industry sectors (e.g., oil-related processes and chemical industry) (Zheng et al., 2018) should be implemented to reduce VOC emissions.

**4 Conclusions**

Surface ozone data from the Chinese Ministry of Environment and Ecology (MEE) network show a sustained nationwide increase over the 2013–2019 period, with a few exceptions (Shandong Province and northeastern China), and with particularly high concentrations in 2018–2019. Correction for meteorologically driven trends with a multiple linear regression (MLR) model shows a general pattern of anthropogenically driven ozone increase across China, though meteorological influences are also significant. The mean summer (JJA) 2013–2019 increase in maximum daily 8-hour

average (MDA8) ozone over China is 1.9 ppb $a^{-1}$ (p<0.01), including 0.7 ppb $a^{-1}$ (p<0.01) from meteorologically driven trends (mostly temperature and circulation) and 1.2 ppb $a^{-1}$ (p<0.01) from anthropogenic influence. Ozone concentrations are highest in the North China Plain (NCP), where the summer mean MDA8 ozone averaged across sites was 83 ppb in 2019 and the summer maximum MDA8 ozone averaged across sites was 129 ppb. In comparison, the Chinese air quality standard for annual maximum MDA8 ozone is 82 ppb. Mean summer MDA8 ozone increased by 3.3 ppb $a^{-1}$ (p<0.01) in

the NCP over the 2013–2019 period, which we attribute as 1.4 ppb $a^{-1}$ (p=0.02) meteorologically driven and 1.9 ppb $a^{-1}$ (p<0.01) anthropogenically driven.

Further investigation of the NCP trends shows that hot weather in June-July 2018–2019 was a major driver for the high ozone concentrations in those summers. Such hot weather does not relate to long-term warming but to interannual variability driven principally by foehn northwesterly winds. Removing this meteorological variability shows a sustained anthropogenic ozone increase over the NCP over the 2013–2019 record and persisting into 2018–2019. Examination of ozone-relevant anthropogenic variables from the MEE network and from satellites shows a 49% decrease in $PM_{2.5}$ for 2013–2019 (15% for 2017–2019), a 25–30% decrease in $NO_x$ emissions for 2013–2019 (6–10% for 2017–2019) and flat VOC emissions. The sustained anthropogenic increase in ozone over the 2017–2019 period may be explained by the continued decrease of $PM_{2.5}$, which scavenges the radical precursors of ozone, combined with flat emissions of VOCs. Reducing VOC emissions should be the top priority for reversing the increase of ozone in the NCP and in other urban areas of China.

*Data availability*. Hourly surface concentrations of air pollutants are archived at http://beijingair.sinaapp.com (last access: 30 June 2020). The MERRA-2 reanalysis data are from http://geoschemdata.computecanada.ca/ExtData/GEOS_0.5x0.625_AS/MERRA2 (last access: 28 February 2020). The L3 OMI satellite data for $NO_2$ and HCHO are available at http://www.qa4ecv.eu/ecvs (last access: 28 February 2020). The L2 TROPOMI data for HCHO are available at https://s5phub.copernicus.eu/dhus (last access: 28 February 2020). The data used in this study can be accessed via doi (https://doi.org/10.7910/DVN/T6D7YY).

*Author contributions*. KL and DJJ designed the study. KL performed the analysis. LS and IDS provided the TROPOMI data. LS, XL, and HL contributed to the interpretation of the results. KL and DJJ wrote the paper with contributions from all co-authors.

*Competing interests*. The authors declare that they have no conflict of interest.

*Acknowledgements*. This work is a contribution from the Harvard-NUIST Joint Laboratory for Air Quality and Climate. HL is supported by the National Natural Science Foundation of China (91744311). We appreciate the efforts from the China Ministry of Ecology and Environment for supporting the nationwide observation network and publishing hourly air pollutant concentrations. We acknowledge the QA4ECV project for the $NO_2$ and HCHO data. We appreciate the efforts from NASA GMAO for providing the MERRA-2 reanalysis data.

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

25

**Table and Figure captions**

**Table 1.** MDA8 ozone trends in China (ppb a$^{-1}$), 2013–2019 and 2013–2017.

**Figure 1.** Summer (JJA) concentrations of maximum MDA8 ozone (**a**), mean MDA8 ozone (**b**), maximum PM$_{2.5}$ (**c**), and mean PM$_{2.5}$ (**d**) for 2013–2019 at the network operated by the China Ministry of Ecology and Environment (MEE). Rectangles denote the four megacity clusters discussed in the text: North China Plain (NCP; 34°–41°N, 113°–119°E), Yangtze River Delta (YRD, 30°–33°N, 119°–122°E), Pearl River Delta (PRD, 21.5°–24°N, 112°–115.5°E), and Sichuan Basin (SCB, 28.5°–31.5°N, 103.5°–107°E).

**Figure 2.** Summertime ozone trends in China, 2013–2019. The left panel (**a**) shows observed trends of summer mean MDA8 ozone at MEE sites averaged on the 0.5° × 0.625° (≈50 × 50 km$^2$) MERRA-2 grid. The trends are obtained by ordinary linear regression and include sites with partial records. The middle panel (**b**) shows meteorologically driven trends determined by fitting ozone to meteorological covariates in the multiple linear regression (MLR) model. The right panel (**c**) shows anthropogenic trends as inferred from the residual of the MLR model. Statistically significant trends above the 90% confidence level are marked with black dots. The mean trends for all of China and for the four megacity clusters (rectangles) are inset, where the regression is applied to the spatially averaged MDA8 ozone for the cluster. Numbers in bold are statistically significant above the 90% confidence level (Table 1).

**Figure 3.** Summer mean trends of 850 hPa wind vectors (m s$^{-1}$ a$^{-1}$) and surface daily maximum temperature (°C a$^{-1}$, shaded) over the period 2013–2019. Data are from the MERRA-2 reanalysis. The trends are obtained by ordinary linear regression of mean JJA data for individual years.

**Figure 4.** Time series of JJA daily maximum surface air temperatures over the North China Plain (NCP) for 1980–2019. Values are monthly means from the MERRA-2 reanalysis. The 2013–2019 period for the ozone trend analysis is shaded in grey.

**Figure 5**. June mean trends in meteorological variables over 2013–2019 under foehn (top) and non-foehn (bottom) conditions (**a**) Trends in 850 hPa winds (m s$^{-1}$ a$^{-1}$) and surface daily maximum temperature (°C a$^{-1}$, shaded) under foehn conditions; (**b**) Trends in 500 hPa winds (m s$^{-1}$ a$^{-1}$) and surface relative humidity (% a$^{-1}$, shaded) under foehn conditions; (**c**, **d**) are the same as (**a**, **b**) but for non-foehn conditions. Data are from the MERRA-2 re-analysis and trends are obtained by ordinary linear regression. Foehn conditions are diagnosed by a foehn index defined by the 850 hPa northwesterly wind averaged along a section from (42°N, 108°E) to (38°N, 112°E) (green line in **a**). The days with positive (negative) foehn index are taken as foehn (no-foehn) conditions. Meteorological data are from the MERRA-2 reanalysis.

**Figure 6.** Trends in summertime ozone and related anthropogenic drivers in the North China Plain (NCP). The left panel (**a**) shows time series of monthly mean MDA8 ozone (ppb) anomalies averaged over the MEE sites relative to the 2013–2019 summer (JJA) mean. Values are shown as anomalies for individual JJA months (3 points per year). Observed trends are compared to the meteorologically driven trends diagnosed by the MLR model, and to the residuals determining the anthropogenically driven trend. The right panel (**b**) shows time series of observed JJA mean quantities averaged over the NCP: $PM_{2.5}$ (black, solid) and $NO_2$ (black, dashed) concentrations from the MEE sites, tropospheric $NO_2$ (blue, solid) and HCHO (pink, solid) column densities from the OMI satellite instrument, and HCHO column density from the TROPOMI satellite instrument (dark blue, solid). Values are presented as ratios relative to 2013. The TROPOMI HCHO data for 2018 have been scaled to the OMI data for that year with the multiplicative factor indicated in legend.

**Table 1.** MDA8 ozone trends in China (ppb a$^{-1}$), 2013–2019 and 2013–2017.

| Regions | JJA 2013–2019 trends | | | JJA 2013–2017 trends | | |
|---|---|---|---|---|---|---|
| | Observed[a] | Meteorological[b] | Anthropogenic[c] | Observed | Meteorological | Anthropogenic |
| China | **1.9** (*<0.01*)[d] | **0.7** (*<0.01*) | **1.2** (*<0.01*) | **1.7** (*<0.01*) | 0.4 (*0.22*) | **1.3** (*<0.01*) |
| NCP | **3.3** (*<0.01*) | **1.4** (*0.02*) | **1.9** (*<0.01*) | **2.7** (*0.01*) | 0.7 (*0.43*) | **2.0** (*<0.01*) |
| YRD | **1.6** (*<0.01*) | 0.7 (*0.12*) | **0.9** (*<0.01*) | **1.7** (*0.03*) | 0.2 (*0.82*) | **1.5** (*<0.01*) |
| PRD | **1.1** (*0.03*) | **0.8** (*0.07*) | 0.3 (*0.29*) | 0.6 (*0.44*) | 0.4 (*0.65*) | 0.3 (*0.51*) |
| SCB | 0.7 (*0.23*) | -0.2 (*0.59*) | **1.0** (*<0.01*) | 0.9 (*0.42*) | 0.1 (*0.90*) | 0.8 (*0.20*) |

[a]Observed trends (OBS) are obtained by ordinary linear regression on summer (JJA) mean values of maximum daily 8-h average (MDA8) ozone measured at the sites of the Ministry of Ecology and Environment (MEE) network. The MDA8 ozone data are first averaged spatially over the $0.5° × 0.625°$ MERRA-2 grid (Figure 2), and then averaged nationally (China) and over four megacity clusters: North China Plan (NCP), Yangtze River Delta (YRD), Pearl River Delta (PRD), Sichuan Basin (SCB).

[b]Meteorologically-driven trends are obtained by fitting the ozone data to a multiple linear regression (MLR) model with the three most important meteorological covariates (see text).

[c]The anthropogenically-driven trends are obtained by ordinary linear regression of the residual ozone after removing the MLR-fitted value.

[d]p-values for the trends are in italics; trends in bold are those with p-value smaller than 0.1.

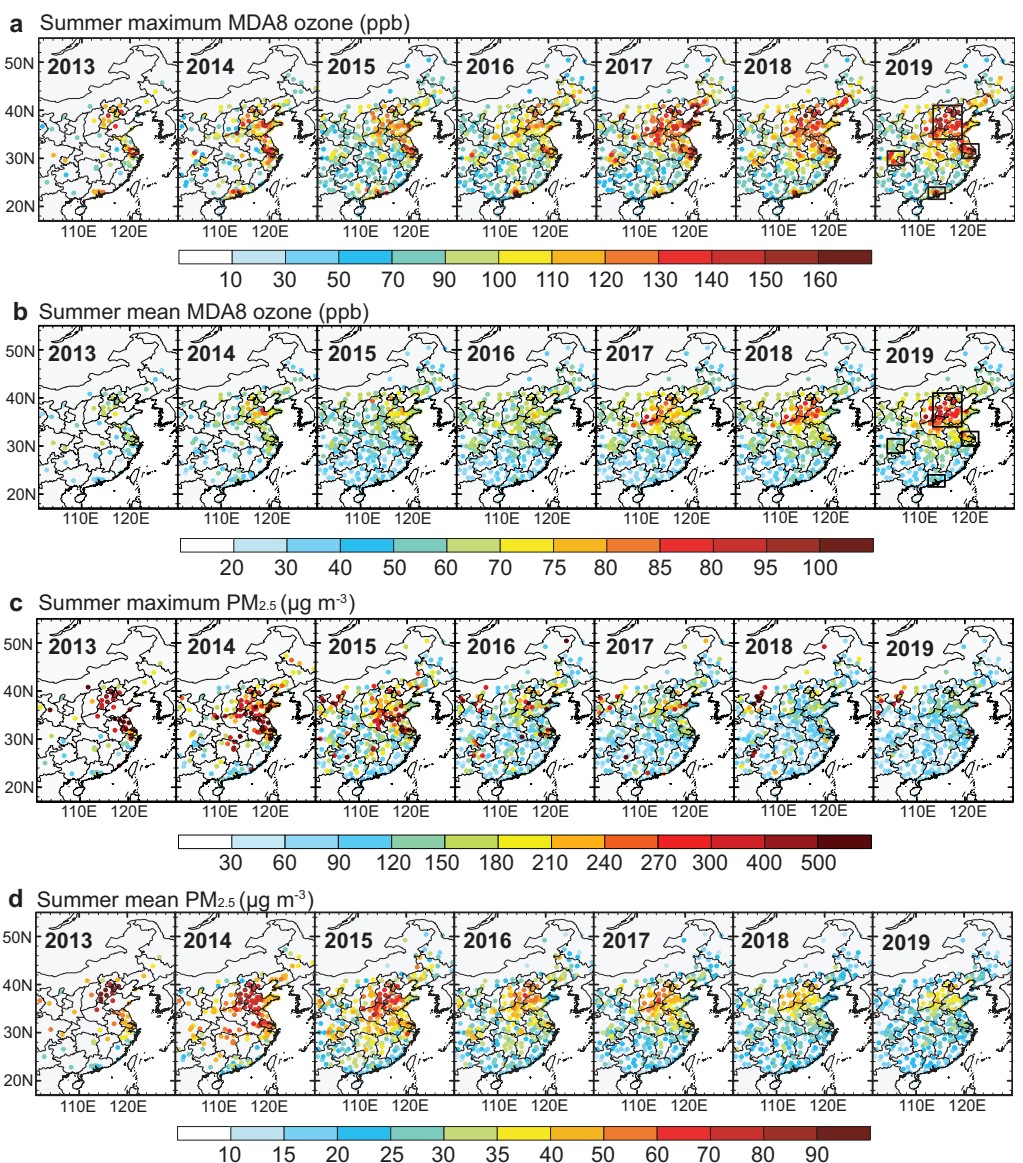

**Figure 1.** Summer (JJA) concentrations of maximum MDA8 ozone (**a**), mean MDA8 ozone (**b**), maximum PM2.5 (**c**), and mean PM2.5 (**d**) for 2013–2019 at the network operated by the China Ministry of Ecology and Environment (MEE). Rectangles denote the four megacity clusters discussed in the text: North China Plain (NCP; 34°–41°N, 113°–119°E), Yangtze River Delta (YRD, 30°–33°N, 119°–122°E), Pearl River Delta (PRD, 21.5°–24°N, 112°–115.5°E), and Sichuan Basin (SCB, 28.5°–31.5°N, 103.5°–107°E).

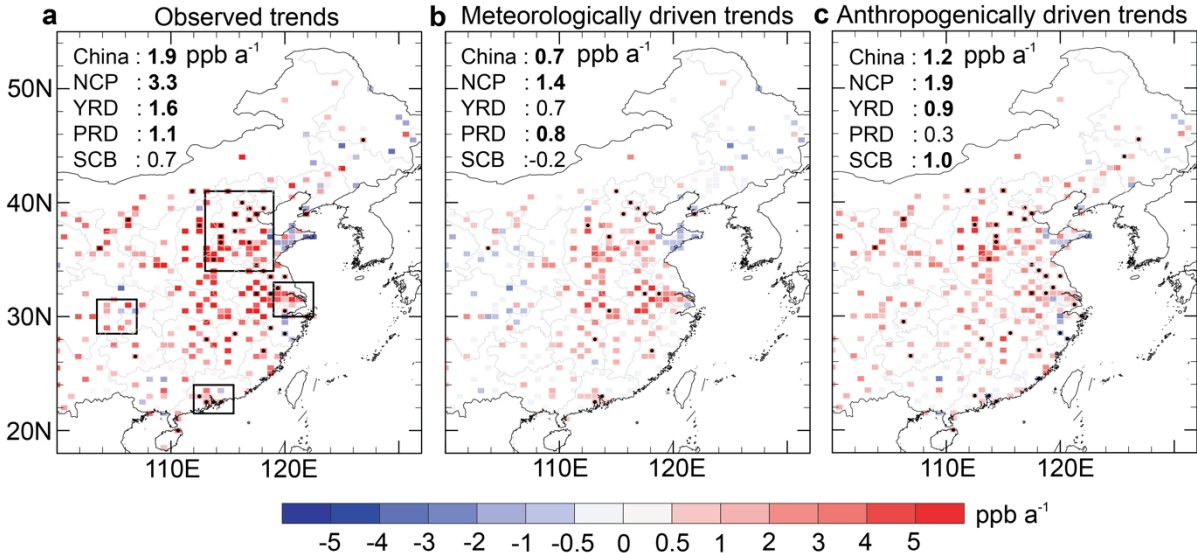

**Figure 2.** Summertime ozone trends in China, 2013–2019. The left panel (**a**) shows observed trends of summer mean MDA8 ozone at MEE sites averaged on the 0.5° × 0.625° (≈50 × 50 km²) MERRA-2 grid. The trends are obtained by ordinary linear regression and include sites with partial records. The middle panel (**b**) shows meteorologically driven trends determined by fitting ozone to meteorological covariates in the multiple linear regression (MLR) model. The right panel (**c**) shows anthropogenic trends as inferred from the residual of the MLR model. Statistically significant trends above the 90% confidence level are marked with black dots. The mean trends for all of China and for the four megacity clusters (rectangles) are inset, where the regression is applied to the spatially averaged MDA8 ozone for the cluster. Numbers in bold are statistically significant above the 90% confidence level (Table 1).

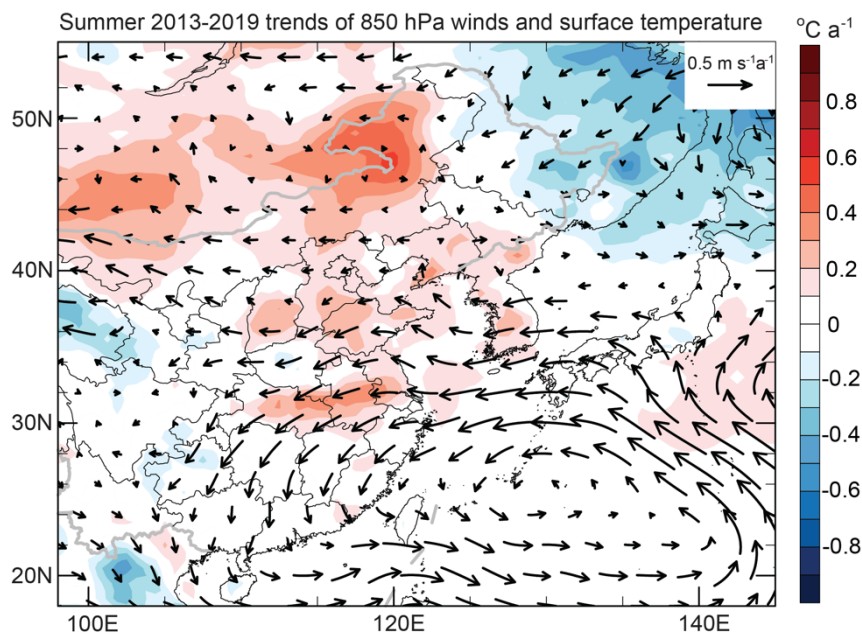

**Figure 3.** Summer mean trends of 850 hPa wind vectors (m s$^{-1}$ a$^{-1}$) and surface daily maximum temperature (°C a$^{-1}$, shaded) over the period 2013–2019. Data are from the MERRA-2 reanalysis. The trends are obtained by ordinary linear regression of mean JJA data for individual years.

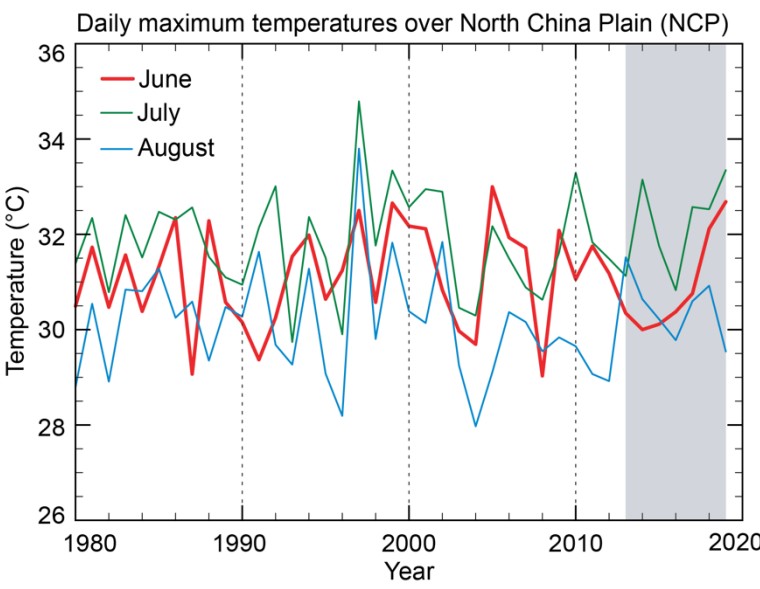

**Figure 4.** Time series of JJA daily maximum surface air temperatures over the North China Plain (NCP) for 1980–2019. Values are monthly means from the MERRA-2 reanalysis. The 2013–2019 period for the ozone trend analysis is shaded in grey.

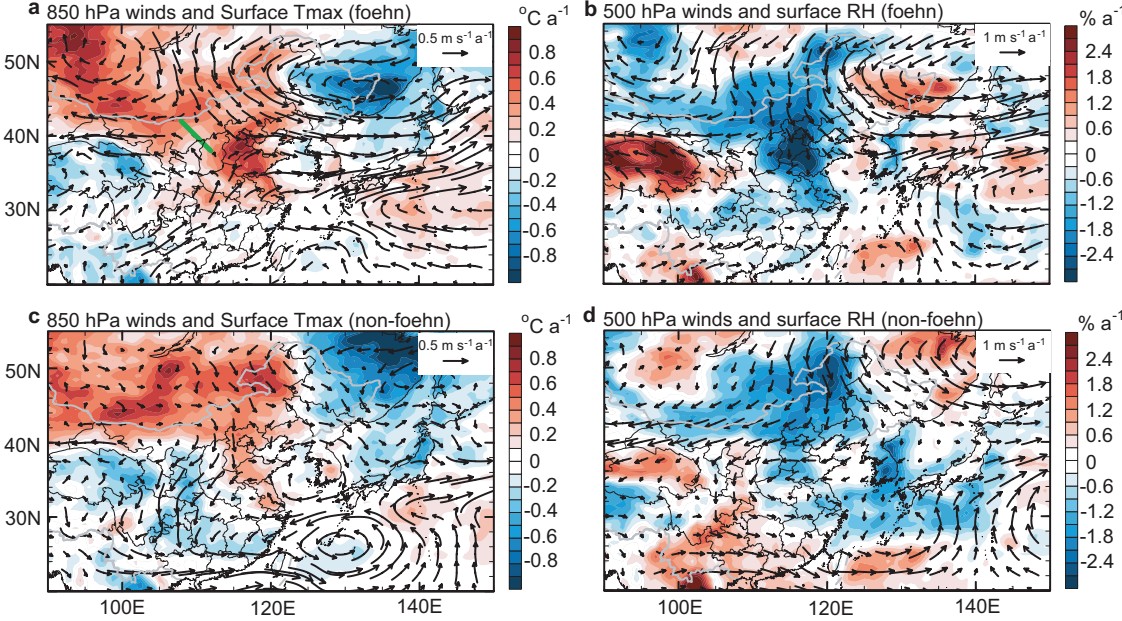

June meteorology trends over 2013-2019 under foehn-favorable (top) and non-foehn (bottom) conditions

**Figure 5**. June mean trends in meteorological variables over 2013–2019 under foehn (top) and non-foehn (bottom) conditions (**a**) Trends in 850 hPa winds (m s$^{-1}$ a$^{-1}$) and surface daily maximum temperature (°C a$^{-1}$, shaded) under foehn conditions; (**b**) Trends in 500 hPa winds (m s$^{-1}$ a$^{-1}$) and surface relative humidity (% a$^{-1}$, shaded) under foehn conditions; (**c**, **d**) are the same as (**a**, **b**) but for non-foehn conditions. Data are from the MERRA-2 re-analysis and trends are obtained by ordinary linear regression. Foehn conditions are diagnosed by a foehn index defined by the 850 hPa northwesterly wind averaged along a section from (42°N, 108°E) to (38°N, 112°E) (green line in **a**). The days with positive (negative) foehn index are taken as foehn (no-foehn) conditions. Meteorological data are from the MERRA-2 reanalysis.

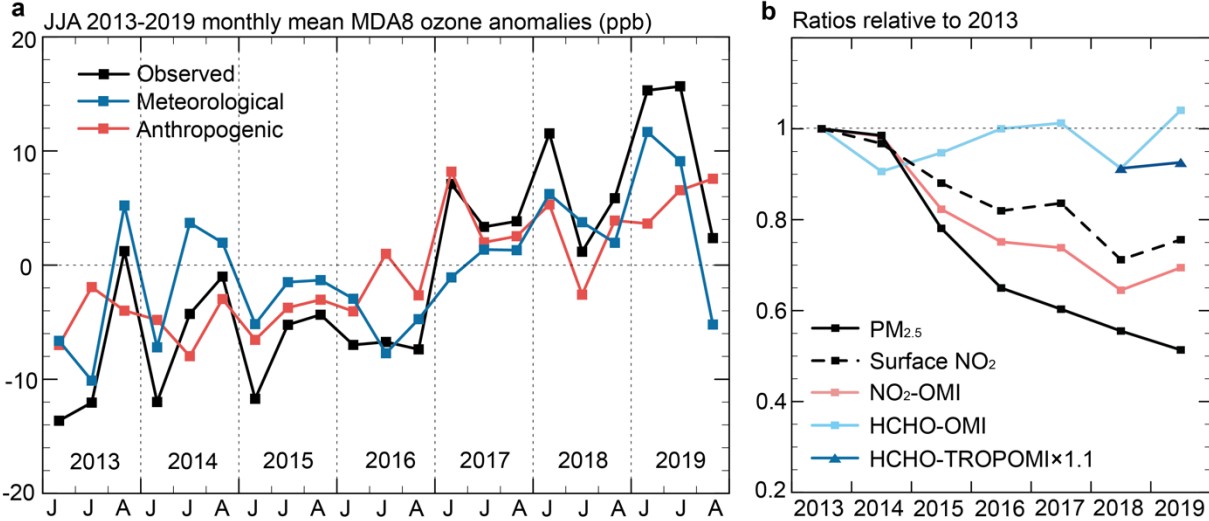

**Figure 6.** Trends in summertime ozone and related anthropogenic drivers in the North China Plain (NCP). The left panel (**a**) shows time series of monthly mean MDA8 ozone (ppb) anomalies averaged over the MEE sites relative to the 2013–2019 summer (JJA) mean. Values are shown as anomalies for individual JJA months (3 points per year). Observed trends are compared to the meteorologically driven trends diagnosed by the MLR model, and to the residuals determining the anthropogenically driven trend. The right panel (**b**) shows time series of observed JJA mean quantities averaged over the NCP: $PM_{2.5}$ (black, solid) and $NO_2$ (black, dashed) concentrations from the MEE sites, tropospheric $NO_2$ (blue, solid) and HCHO (pink, solid) column densities from the OMI satellite instrument, and HCHO column density from the TROPOMI satellite instrument (dark blue, solid). Values are presented as ratios relative to 2013. The TROPOMI HCHO data for 2018 have been scaled to the OMI data for that year with the multiplicative factor indicated in legend.