# Peer review of "2013–2019 increases of surface ozone pollution in China: anthropogenic and meteorological influences"

_Atmospheric Chemistry and Physics, 2020_

## Referee Comment (RC1) · Anonymous Referee #1 · 16 May 2020

Review of "2013-2019 increases of surface ozone pollution in China: anthropogenic and meteorological influences" by Ke Li, Daniel J. Jacob, Lu Shen, Xiao Lu, Isabelle De Smedt and Hong Liao, submitted to Atmospheric Chemistry and Physics.

Summary:

Based on statistical analysis of recent surface measurements of ozone in China and meteorological conditions from a modern reanalysis product, the authors quantify the role of meteorology versus emissions on the positive ozone trends in Chinese megacities. Despite regulations targeting ozone pollution, ozone has continued to increase over the past decade.

While the results are interesting, they seem preliminary. Section 3 is a list of "Figure X shows" and I encourage the authors to add depth to their research and analysis and more fully develop their narrative. There are five figures, which are all multi-paneled, and only three pages of "Results and discussion"; this unequal balance of figure to text highlights the need for further exploration into the information contained in each figure. In addition, there is a supplemental figure and supplemental table which I found added to the analysis and I encourage the authors to include these in the main text.

This manuscript is within the scope for Atmospheric Chemistry and Dynamics; however, I suggest the authors expand the discussion of the results before I recommend it for publication.

**Major comments:**

Pg 2 Ln 15: The authors discuss VOCs as additional industrial sources but there can be natural sources of VOCs from plants. Have the authors considered the natural sources of VOCs in this analysis? There's no discussion of natural VOC emissions on Page 6 or 7.

Pg2 Ln 27: The authors should include the MEE website here in the body of the text and in the Data Availability section at the end of the manuscript.

Pg 4 Ln 15, 20: These trend values are provided in Figure 2. What is novel about Table S1 is it highlights that a significance test was performed. In Line 20, the authors quote "significantly enhanced" without a reference to a statistical test. The authors should describe this in the Data and Methods Section at the bottom of Page 3 and I would encourage the authors to include this table in the main text as it is referenced on Page 4 Line 21 to support a critical result.

Pg4 Ln 30: The authors reference a supplemental figure which I would argue should be in the main text as it highlights not only the change in maritime inflow which impacts the Shandong Peninsula, but it looks like the YRD and Northeast China as well. The authors should discuss how the meteorological conditions mitigate ozone pollution over western China (does that include SCB?) and northeast China.

Pg4 Lines 22,23 and Page 5 Ln15: The authors could go into more detail on not only the primary meteorological predictor variables, and also include the breakdown for all megacities and regions discussed. Only NCP and PRD's principal predictors are given.

**Figure comments:**

Pg 4 Line 7: It would make it easier on the reader if the rectangles in 2019 Figure 1 top row where included in 2019 Figure 1 middle row since the text is discussing both mean MDA8 and the max MDA8 by region.

Pg 14 Figure 1:

  While the figure caption includes the latitude and longitude for the four megacity clusters, the latitude and longitude ticks are not labelled. Can [some of] the ticks be labelled, or at least include in the caption what are the intervals of the major and minor ticks and some reference point?

  Would the discussion of the ozone max and mean trends benefit from the max PM2.5 being included in Figure 1? Has the maximum PM2.5 decreased the same as the mean (Pg4 Lines 4-5). If that is the case, could state that and not show it.

Page 15 Figure 2: Can the rectangles for the megacity clusters be added to Figure 2 or at least Figure 2a? I did my best to draw them on so I could follow the text referring to the trends in the four regions. Again, it would be helpful to have some of the latitude and longitude tick marks labelled and/or the intervals of the major and minor ticks and some reference point defined in the figure caption.

Page 15 Line 10: Could add a reference to the Table S1 at the end of the caption.

Pg 5 Line 8: This sentence references the Table in Figure 3. The left figure and right table should be labelled as (a) and (b) to make referencing in the text clearer. Also, to save on white space, I would encourage the authors to include the table as an inset in Figure 3 or as a separate table.

Pg 17 Line 7-9: The definition of the Foehn index and foehn-favorable conditions should be in Section 2 or in the text of Section 3; I suggest it is removed from the figure caption.
Pg 17 Line 9-10: "The frequency of foehn wind under hot days increased by 85% over the period" is a result and should be in the main text and not in the figure caption. Can the authors go into more detail about this trend? Was it mainly driven by 2018 and 2019?

**Minor and technical comments:**

Pg1Ln19: The June-July temperatures over the NCP are higher than what? Other regions of China? Other months?

Pg2Ln18: Can the authors describe how meteorological conditions may affect emissions? Are they referring to natural emissions from plants which do vary based on meteorological conditions, or do they mean anthropogenic emissions such as through energy consumption?

Pg 3 Line 1: No mention of NO2 surface observations but these are referenced in Figure 5.

Pg 3 Line 12: There was a version change in the TROPOMI NO2 data in March 2019 (https://sentinel.esa.int/documents/247904/2474726/Sentinel-5P-Level-2-Product-User-Manual-Nitrogen-Dioxide).  I am concerned this change could add a bias when comparing Summer 2018 (v1.2.0) vs Summer 2019 (v1.3.0).

Pg 4 Line 2: A paragraph of preamble providing an overview of the results section would be good, to help the reader see that there are three subsections within Section 3.

Pg 4 Line 10: Do the authors have a hypothesis as to why the summer maximum MDA8 ozone values in YRD, PRD, and SCB were comparable to the NCP but not the means?

Pg 4 Ln 14: Can the authors provide latitude and longitude regions for Shandong Peninsula and Northeast China.  Line 28, the authors refer to 'northeastern' China.  Is this different than Northeast China?

Pg 5 Lines 2-3: It looks to me that the anthropogenic trend is more uniformly positive in Fig 2c than Figs 2a,b except for in the Shandong Peninsula and maybe the PRD and YRD regions.  Can the authors confirm?

Pg 5 Line 4: Why might the PRD experience a decrease?  Is that because of the change in monsoon winds?  Make connections between the figures and discussion where possible.

Pg 5 Line 16: While the June (August) temperatures clearly show increasing (decreasing) trends over the 2013-2019 period, the temperature pattern in July looks almost neutral if averaged over this period and not "a lesser increase".  This phrase is awkward.

Pg 5 Line 21:  The reference following foehn winds gives me the impression that this paper is the first to define foehn winds.  However, a foehn wind is the warming of air through adiabatic descent on the lee side of a mountain, much like a Chinook Wind on the lee side of the Rocky Mountains, so the reference is likely more appropriate at the end of the sentence.  The authors should describe the foehn wind in meteorological terms, whereby air which is forced to rise over the mountains, loses much of its water vapor to condensation on the windward side, and subsequently warms dry adiabatically as it descends on the lee side.

Pg 5 Line 22,23: The phrase "winds blow from the mountains to the north and west" is confusing.  Either the mountains are to the north and west of NCP and the wind blows from the mountains to the NCP, or the wind blows to the north and west, from the mountains to NCP. Possibly this level of detail could be included in the meteorological definition for the foehn wind which would simplify this sentence, or the use of latitude and longitude for the mountain range versus the region defined for the NCP.

Pg 6 Line 3: Are the MEE sites in the NCP average all full time series or are any partial records during the period? It would be good to state that like you did for Figure 2.

Pg 6 Line 6: Can the authors quantify the "much less month-to-month variability" (e.g., possibly through the standard deviation)?

Pg 6 Line 9,15: Provide a reference that NO2 is a proxy for NOx emissions. Could instead include this idea and a reference in the introduction (Pg 2).

Pg 6 Line 14-15: The authors quote decreases in PM2.5 and NOx emissions for 2017-2019 but the base year or period is not provided.

Pg 6 Line 28: add "and" between "Province and Northeast"

Pg 7 Line 5-6: How do these values compare to the Chinese and US National Ambient Air Quality Standards? Good to put this in perspective of the health standards.

Pg 7 Line 9: Change "warning" to "warming"

---

## Referee Comment (RC2) · Anonymous Referee #2 · 13 Jun 2020

This study quantifies the most recent trends in summertime O3 concentrations in China and investigated the possible causes. This is a timely paper which has implications for the improvement of China's ongoing control policies. However, I have the following concerns which need to be addressed before the manuscript can be considered for publication in ACP.

Major comments:

1. The multiple linear regression (MLR) is a key method used in this study to quantify the meteorological contribution. However, this paper lacks a lot of details regarding the data sources and results of the MLR method. In Section 2: "The regression model is first applied to select the key meteorological parameters driving the day-to-day variability of ozone for each grid cell." What meteorological parameters are considered in

the selection? Which parameters show statistically significant contribution based on the regression? What criteria did you use the select the parameters used in the formal analysis? How much did the selected parameters explain the overall variability? In Section 3.1 and Section 3.2, you talked a lot about the dominant meteorological predictors in China and various metropolitan regions. However, no MLR results supporting these conclusions are shown. How much did these parameters contribute? Are the contributions from these parameters statistically significant?

2. After reading the paper, my overall impression is that the author should tune down the statement that they have elucidated the relative contribution of meteorological and anthropogenic factors to the O3 trend. The meteorologically driven trend is quantified by fitting O3 to selected met parameters while the residual is regarded as the anthropogenically driven trend, so the anthropogenically driven trend is largely unconstrained. This attribution method is subject to a large uncertainty, especially for the anthropogenically driven part. I would not recommend the author to conduct a modeling simulation to test the anthropogenic contribution which requires a lot of additional work, but I am deeply concern that the quantitative attribution to the two parts may not be accurate without further constraint. Even for the meteorological part, you only considered a subset of met parameters in the MLR. Can these selected parameters represent the overall contribution of meteorology? This again points to my last comment that showing the results of the MLR analysis is important.

3. Section 3.1: When you talk about the observational trends, you need to point out whether these trends are statistically significant. Fig. 2 shows some significance testing results, but it's also important to incorporate such information in your description.

4. Abstract Line 20-22: Whether the anthropogenically driven O3 trend is caused by decrease in PM2.5 or reduction in NOx is a controversial issue. This study actually did not carefully investigate this issue but just referred to a previous study. Therefore, you may at most infer that this might be a cause rather than state with certainty that this is the actual explanation.

5. In your regression analysis to determine the O3 trend, you included sites with partial records. Since the number of observational sites grow dramatically from 2013 to 2019, the trends can be biased by the differences in observational sites. I suggest that you repeat the analysis using only continuous sites and examine whether this affects your results significantly.

Minor comments:

1. Sometimes you abbreviated "meteorologically driven trends" to "meteorological trends", which I think is not accurate.

2. The spatial extents of NCP, YRD, PRD, and SCB are not defined in the paper.

---

## Author Comment (AC1) · 25 Jul 2020

**Response to Review #1**

Review of "2013-2019 increases of surface ozone pollution in China: anthropogenic and meteorological influences" by Ke Li, Daniel J. Jacob, Lu Shen, Xiao Lu, Isabelle De Smedt and Hong Liao, submitted to Atmospheric Chemistry and Physics.

Summary: Based on statistical analysis of recent surface measurements of ozone in China and meteorological conditions from a modern reanalysis product, the authors quantify the role of meteorology versus emissions on the positive ozone trends in Chinese megacities. Despite regulations targeting ozone pollution, ozone has continued to increase over the past decade.

While the results are interesting, they seem preliminary. Section 3 is a list of "Figure X shows" and I encourage the authors to add depth to their research and analysis and more fully develop their narrative. There are five figures, which are all multi-paneled, and only three pages of "Results and discussion"; this unequal balance of figure to text highlights the need for further exploration into the information contained in each figure. In addition, there is a supplemental figure and supplemental table which I found added to the analysis and I encourage the authors to include these in the main text.

This manuscript is within the scope for Atmospheric Chemistry and Dynamics; however, I suggest the authors expand the discussion of the results before I recommend it for publication.

We appreciate the reviewer's constructive and thorough comments/suggestions. We have moved the supplementary figure and table to the main text. We also have detailed the description of MLR model and foehn wind effect. Figures have been also revised following your suggestions, thanks! Please find below our point-by-point response in **blue**.

**Major comments**:

Pg 2 Ln 15: The authors discuss VOCs as additional industrial sources but there can be natural sources of VOCs from plants. Have the authors considered the natural sources of VOCs in this analysis? There's no discussion of natural VOC emissions on Page 6 or 7.

Thanks. We have added the discussion in P4L28-30: "The effect of biogenic VOCs on ozone trends depends on meteorological and land cover drivers. Meteorological drivers, in particular temperature, would be accounted for in the MLR model. The effect of land cover changes is expected to be small over the 7-year time horizon of our analysis (Fu and Tai, 2014)."

Pg2 Ln 27: The authors should include the MEE website here in the body of the text and in the Data Availability section at the end of the manuscript.
Added.

Pg 4 Ln 15, 20: These trend values are provided in Figure 2. What is novel about Table S1 is it highlights that a significance test was performed. In Line 20, the authors quote "significantly enhanced" without a reference to a statistical test. The authors should describe this in the Data and Methods Section at the bottom of Page 3 and I would encourage the authors to include this table in the main text as it is referenced on Page 4 Line 21 to support a critical result.

Thanks! Now it (as Table 1) has been added in the main text. For statistical test, we have added in P4L24-25: "with the statistical significance of the anthropogenic trend determined by Student's *t*-test"

Pg4 Ln 30: The authors reference a supplemental figure which I would argue should be in the main text as it highlights not only the change in maritime inflow which impacts the Shandong Peninsula, but it looks like the YRD and Northeast China as well. The authors should discuss how the meteorological conditions mitigate ozone pollution over western China (does that include SCB?) and northeast China.

Thanks! Now it (as Figure 3) has been added in the main text. We also have added the discussion on the decreasing ozone trends in P6L8-9: "Temperature decreased over northeastern China (**Figure 3**)."

Pg4 Lines 22,23 and Page 5 Ln15: The authors could go into more detail on not only the primary meteorological predictor variables, and also include the breakdown for all megacities and regions discussed. Only NCP and PRD's principal predictors are given.

We have added the plots of leading meteorological variables in Supplementary Figures S1-3, which are also cited in the main text.

**Figure comments**:
Pg 4 Line 7: It would make it easier on the reader if the rectangles in 2019 Figure 1 top row where included in 2019 Figure 1 middle row since the text is discussing both mean MDA8 and the max MDA8 by region.
Added.

Pg 14 Figure 1:
   While the figure caption includes the latitude and longitude for the four megacity clusters, the latitude and longitude ticks are not labelled. Can [some of] the ticks be labelled, or at least include in the caption what are the intervals of the major and minor ticks and some reference point?
   Would the discussion of the ozone max and mean trends benefit from the max PM2.5 being included in Figure 1? Has the maximum PM2.5 decreased the same as the mean (Pg4 Lines 4-5). If that is the case, could state that and not show it.

We have revised Figure 1, and the maximum $PM_{2.5}$ trends are also added to Figure 1. We have added in P5L9-10 about the decrease of maximum $PM_{2.5}$: "Maximum $PM_{2.5}$ concentrations experienced a similar decrease trend."

Page 15 Figure 2: Can the rectangles for the megacity clusters be added to Figure 2 or at least Figure 2a? I did my best to draw them on so I could follow the text referring to the trends in the four regions. Again, it would be helpful to have some of the latitude and longitude tick marks labelled and/or the intervals of the major and minor ticks and some reference point defined in the figure caption.
Done.

Page 15 Line 10: Could add a reference to the Table S1 at the end of the caption.

Added.

Pg 5 Line 8: This sentence references the Table in Figure 3. The left figure and right table should be labelled as (a) and (b) to make referencing in the text clearer. Also, to save on white space, I would encourage the authors to include the table as an inset in Figure 3 or as a separate table.

The ozone trend has been already given in the main text and We have removed the right table now.

Pg 17 Line 7-9: The definition of the Foehn index and foehn-favorable conditions should be in Section 2 or in the text of Section 3; I suggest it is removed from the figure caption.

Thanks! We have included the definition of foehn index in P7L-10 in Section 3. We would like to still include this in the Figure caption to make the Figure self-explanatory.

Pg 17 Line 9-10: "The frequency of foehn wind under hot days increased by 85% over the period" is a result and should be in the main text and not in the figure caption. Can the authors go into more detail about this trend? Was it mainly driven by 2018 and 2019?

Yes. We have moved this into the main text in P7L11-13: "The frequency of foehn conditions under hot days in June increased by 85% over the 2013–2019 period (driven mainly by the increased frequency in 2018–2019), and ozone increase under foehn conditions is 1.2 ppb $a^{-1}$ larger than under no-foehn conditions."

**Minor and technical comments**:

Pg1Ln19: The June-July temperatures over the NCP are higher than what? Other regions of China? Other months?

Higher than previous years. We have changed "higher" to "rising" for clarification.

Pg2Ln18: Can the authors describe how meteorological conditions may affect emissions? Are they referring to natural emissions from plants which do vary based on meteorological conditions, or do they mean anthropogenic emissions such as through energy consumption?

We intended to say "natural emissions", which are much more meteorologically dependent than anthropogenic emissions. We have added "natural" in P2L19.

Pg 3 Line 1: No mention of NO2 surface observations but these are referenced in Figure 5.
Added.

Pg 3 Line 12: There was a version change in the TROPOMI NO2 data in March 2019 (https://sentinel.esa.int/documents/247904/2474726/Sentinel-5P-Level-2-Product-UserManual-Nitrogen-Dioxide). I am concerned this change could add a bias when comparing Summer 2018 (v1.2.0) vs Summer 2019 (v1.3.0).

Thank you for pointing this out. Now we have removed the results for TROPOMI $NO_2$ changes.

Pg 4 Line 2: A paragraph of preamble providing an overview of the results section would be good, to help the reader see that there are three subsections within Section 3.

Great suggestion.
We have added in this in the beginning of Section 3: "We first present the general 2013–2019 summer ozone trends in China and their statistically decomposed meteorological and anthropogenic contributions. Ozone trends over the major megacity clusters in China are highlighted. We go on to more specifically attribute the meteorological and anthropogenic drivers of recent ozone trends over the North China Plain, where the ozone increase is the highest.".

Pg 4 Line 10: Do the authors have a hypothesis as to why the summer maximum MDA8 ozone values in YRD, PRD, and SCB were comparable to the NCP but not the means?

We have added the explanation in P5L16-19: "These three megacity clusters are subject to similar ozone pollution episodes under stagnant conditions as the NCP (Wang et al., 2017), but they are more frequently ventilated by the summer monsoon bringing cleaner tropical air and precipitation hence the lower mean ozone.".

Pg 4 Ln 14: Can the authors provide latitude and longitude regions for Shandong Peninsula and Northeast China. Line 28, the authors refer to 'northeastern' China. Is this different than Northeast China?

We now use "northeastern" instead of "Northeast" throughout the text for consistency. We intended to focus on the four major megacities clusters that are also the pollution control regions targeted by the Chinese government.

Shandong Peninsula typically refers to the east part of Shandong province; Northeast China typically includes Helongjiang, Jilin, and Liaoning provinces. We have added this information in the main text (P5L22).

Pg 5 Lines 2-3: It looks to me that the anthropogenic trend is more uniformly positive in Fig 2c than Figs 2a,b except for in the Shandong Peninsula and maybe the PRD and YRD regions. Can the authors confirm?

Yes, you are right. The meteorological role varies regionally. We have revised the text in P6L11-12: "This anthropogenic trend is more uniformly positive at a national scale…".

Pg 5 Line 4: Why might the PRD experience a decrease? Is that because of the change in monsoon winds? Make connections between the figures and discussion where possible.

Yes, due to weakened monsoonal winds. We have added the explanation in P6L15-16: "The ozone increase in PRD is mainly meteorologically driven due to reduced monsoonal winds (**Figure 3**).".

Pg 5 Line 16: While the June (August) temperatures clearly show increasing (decreasing) trends over the 2013-2019 period, the temperature pattern in July looks almost neutral if averaged over this period and not "a lesser increase". This phrase is awkward.

In fact, the temperature trend in July (0.22 °C a$^{-1}$) is comparable with trend in August (–0.18 °C a$^{-1}$), while both of them are much lower than trend in June (0.42 °C a$^{-1}$). The high temperature in 2018–2019 is the reason for an increasing temperature trend in July. We have added the temperature trends in P6L29 and P7L1.

Pg 5 Line 21: The reference following foehn winds gives me the impression that this paper is the first to define foehn winds. However, a foehn wind is the warming of air through adiabatic descent on the lee side of a mountain, much like a Chinook Wind on the lee side of the Rocky Mountains, so the reference is likely more appropriate at the end of the sentence. The authors should describe the foehn wind in meteorological terms, whereby air which is forced to rise over the mountains, loses much of its water vapor to condensation on the windward side, and subsequently warms dry adiabatically as it descends on the lee side.

Great suggestion.
We have added the introduction in P7L5-7: "Foehn wind conditions featuring warm and dry air subsiding from the mountains that are to the north and west of the NCP (Chen and Lu, 2016) also lead to high ozone pollution in the NCP."

Pg 5 Line 22,23: The phrase "winds blow from the mountains to the north and west" is confusing. Either the mountains are to the north and west of NCP and the wind blows from the mountains to the NCP, or the wind blows to the north and west, from the mountains to NCP. Possibly this level of detail could be included in the meteorological definition for the foehn wind which would simplify this sentence, or the use of latitude and longitude for the mountain range versus the region defined for the NCP.

Revised. Please see our response to last comment.

Pg 6 Line 3: Are the MEE sites in the NCP average all full time series or are any partial records during the period? It would be good to state that like you did for Figure 2.

We have revised in P7L17: "…all MEE sites in the NCP and including sites with partial records."

Pg 6 Line 6: Can the authors quantify the "much less month-to-month variability" (e.g., possibly through the standard deviation)?

Thanks! We have added this information in P7L21-22: "The standard deviation decreases from 8.8 ppb to 5.3 ppb after removal of meteorological influence.".

Pg 6 Line 9,15: Provide a reference that NO2 is a proxy for NOx emissions. Could instead include this idea and a reference in the introduction (Pg 2).

Added. We also revised the text in P2L26-28: "bring in satellite and ground-based observations to relate the most recent ozone trends to those of VOC (Shen et al., 2019b) and $NO_x$ emissions (Zheng et al., 2018; Shah et al., 2020)."

Pg 6 Line 14-15: The authors quote decreases in PM2.5 and NOx emissions for 2017-2019 but the base year or period is not provided.

Base year is 2017. We have revised in P8L2: "Relative to 2017, we find for 2019 a 15% …"

Pg 6 Line 28: add "and" between "Province and Northeast"
Done.

Pg 7 Line 5-6: How do these values compare to the Chinese and US National Ambient Air Quality Standards? Good to put this in perspective of the health standards.

We have added in P8L23-24: "In comparison, the Chinese air quality standard for annual maximum MDA8 ozone is 82 ppb."

Pg 7 Line 9: Change "warning" to "warming"
Changed.

---

## Author Comment (AC2) · 25 Jul 2020

**Response to Review #2**

This study quantifies the most recent trends in summertime O3 concentrations in China and investigated the possible causes. This is a timely paper which has implications for the improvement of China's ongoing control policies. However, I have the following concerns which need to be addressed before the manuscript can be considered for publication in ACP.

We thank the reviewer's valuable comments which improve our manuscript greatly. We have detailed the MLR method and justified the application of this statistical approach. Please find below our point-by-point response in **blue**.

Major comments:

1. The multiple linear regression (MLR) is a key method used in this study to quantify the meteorological contribution. However, this paper lacks a lot of details regarding the data sources and results of the MLR method. In Section 2: "The regression model is first applied to select the key meteorological parameters driving the day-to-day variability of ozone for each grid cell." What meteorological parameters are considered in the selection? Which parameters show statistically significant contribution based on the regression? What criteria did you use the select the parameters used in the formal analysis? How much did the selected parameters explain the overall variability? In Section 3.1 and Section 3.2, you talked a lot about the dominant meteorological predictors in China and various metropolitan regions. However, no MLR results supporting these conclusions are shown. How much did these parameters contribute? Are the contributions from these parameters statistically significant?

Sorry for not making it clear. The statistical method follows our previous study (Li et al., 2019a). We now have detailed the MLR method in P4L5-21 in Section 2, and have included the meteorological candidates to be selected in the regression model, how the top three meteorological drivers are selected for each grid cell, and the explained variance by the MLR model.

"Firstly, the regression model is applied to select the key meteorological parameters driving the day-to-day variability of ozone for each grid cell. There are nine MERRA-2 meteorological variables considered as ozone covariates, including daily maximum 2-m air temperature (Tmax), 10-m zonal wind (U10) and meridional wind (V10), boundary layer height (PBLH), total cloud area fraction (TCC), rainfall (Rain), sea level pressure (SLP), relative humidity (RH), and 850-hPa meridional wind (V850), following (Li et al., 2019a). The meteorology fields are averaged over 24 h for use in the MLR model except for PBLH and TCC, which are averaged over daytime hours (8–20 local time), and for Tmax (daily maximum).

Secondly, to avoid overfitting, only the three locally dominant meteorological parameters are regressed onto the deseasonalized monthly MDA8 ozone to fit the role of 2013–2019 meteorological variability. The top three variables are selected based on their individual contribution to the regressed ozone, along with the requirement that they are statistically significant above the 95% confidence level in the MLR model. They will differ for each 0.5° × 0.625° grid cell. We show these top three meteorological drivers for ozone variability in Figure S1–S3 for different locations in China.

Thirdly, we fit the observed monthly ozone anomalies by applying these dominant meteorological drivers in the MLR model. The coefficients of determination (R2) for the MLR model are generally above 0.4–0.5 for polluted regions of China which are of most interest to us (Figure S4). Remote locations with background ozone levels have less ozone variability and are thus harder to fit.".

2. After reading the paper, my overall impression is that the author should tune down the statement that they have elucidated the relative contribution of meteorological and anthropogenic factors to the O3 trend. The meteorologically driven trend is quantified by fitting O3 to selected met parameters while the residual is regarded as the anthropogenically driven trend, so the anthropogenically driven trend is largely unconstrained. This attribution method is subject to a large uncertainty, especially for the anthropogenically driven part. I would not recommend the author to conduct a modeling simulation to test the anthropogenic contribution which requires a lot of additional work, but I am deeply concern that the quantitative attribution to the two parts may not be accurate without further constraint. Even for the meteorological part, you only considered a subset of met parameters in the MLR. Can these selected parameters represent the overall contribution of meteorology? This again points to my last comment that showing the results of the MLR analysis is important.

The application of MLR mode has been detailed in our response to the last comment. This method has been extensively applied to quantify the effect of meteorological variability on air pollutants, and statistical quantification of anthropogenic and meteorological contributions to air pollutants also has been well documented. We have clarified this in the main text.

In P4L21-22: "Similar MLR models have been extensively employed to quantify the effect of meteorological variability on air pollutants (e.g., Tai et al., 2010; Otero et al., 2018; Zhai et al., 2019; Han et al., 2020).".

In P4L25-28: "We have followed this approach before to isolate the anthropogenic trends of ozone and $PM_{2.5}$ (Li et al., 2019a; Zhai et al., 2019). Similar statistical decomposition of anthropogenic and meteorological contributions to air pollutant trends has been also employed by previous studies (e.g., Chen et al. 2019; Yu et al., 2019; X. Zhang et al., 2019)."

3. Section 3.1: When you talk about the observational trends, you need to point out whether these trends are statistically significant. Fig. 2 shows some significance testing results, but it's also important to incorporate such information in your description.

We have moved Table S1 into the main text, as also suggested by Reviewer #1. A p-value showing significance is also given wherever applicable.

4. Abstract Line 20-22: Whether the anthropogenically driven O3 trend is caused by decrease in PM2.5 or reduction in NOx is a controversial issue. This study actually did not carefully investigate this issue but just referred to a previous study. Therefore, you may at most infer that this might be a cause rather than state with certainty that this is the actual explanation.

Thanks. We agree with the reviewer. We have revised the text accordingly.

In P1L21: "fine particulate matter ($PM_{2.5}$) that may be driving the continued anthropogenic increase in ozone"
In P5L24-26: "The increases are largest in the NCP, which could be explained by greater influence of radical scavenging by $PM_{2.5}$ (Li et al., 2019a, 2019b)."
In P9L3-4: "The sustained anthropogenic increase in ozone over the 2017–2019 period may be explained by the continued decrease of $PM_{2.5...}$".

5. In your regression analysis to determine the O3 trend, you included sites with partial records. Since the number of observational sites grow dramatically from 2013 to 2019, the trends can be biased by the differences in observational sites. I suggest that you repeat the analysis using only continuous sites and examine whether this affects your results significantly.

The sites are basically stable after 2014. Our results still stand if only continuous records are used, as shown in the following plot.

We have added the plot in the Supplementary Information, and description in main text P6L15: "This result still stands if only continuous records since 2013 are used in the analysis (Figure S5)."

[Figure]

**Figure S5.** Same with Figure 2 but for the sites with continuous records from 2013.

Minor comments:
1. Sometimes you abbreviated "meteorologically driven trends" to "meteorological trends", which I think is not accurate.

Corrected throughout the text. Thanks.

2. The spatial extents of NCP, YRD, PRD, and SCB are not defined in the paper

Added in P5L12-13.

---

## Referee Report (RR1)

Second Review of "2013-2019 increases of surface ozone pollution in China: anthropogenic and meteorological influences" by Li et al. submitted to Atmospheric Chemistry and Physics.

The authors did an excellent job addressing my comments from the initial submission. I have a few minor (mostly technical) edits below and I recommend the paper for publication once these have been addressed. One thing that hit me as I read toward the end of the paper was whether or not the authors have looked into the emission sectors. They suggest that VOC emissions need to be reduced but are these coming from a different sector than the NOx emissions (and what has led to a reduction in PM2.5). It would be interesting to note it at the end of Section 3.3 or in Section 4 Conclusions (Pg 9 Lines 1-6) if there was more guidance on where targeted efforts need to be made to reduce VOC emissions.

Minor comments:
Pg 1 Line 20-22: Suggest changing "flat" to "constant" and maybe removing the comma after ozone. The "and flat emissions of volatile organic compounds" seems disconnected from the "NCP data show" at the start of the sentence.
Pg 2 Line 16: Space before "OH"
Pg 3 Line 18: I think remove the "for". It ends as "2018-2019 for (De Smedt et al., 2018)." and I think that "for" is left over.
Pg 4 Line 3-11; Pg 9 Line 8: Each MERRA-2 data collection has its own DOI that can be referenced. The website address to the MERRA-2 webpage is not where you accessed the data, and the Gelaro et al. paper is sufficient a reference for MERRA-2 on Page 4. If you downloaded the MERRA-2 data yourself, you likely downloaded from the GES DISC (https://disc.gsfc.nasa.gov) and there are details there on how to properly cite the data in publications and that is what should be referenced in the text and in the Data availability section.
Pg 4 Line 27: remove "also"
Pg 5 Line 8: Remove one of the two instances of "mean" (likely the first).
Pg 5 Line 12: a semicolon is used after NCP while only a comma after the other three region abbreviations.
Pg 5: Line 18: Are you sure the mean is not lower because of lower ozone from NOx titration?
Pg 6: Line 7-9: Can the last two sentences focus on the results from Figure 3 and not stress that a finding is seen in a supplemental figure "As shown in Figure S2".
Pg 6: Line 14-15: Since sentence starts out with ozone increases in all the megacity clusters, why is only the NCP trend highlighted at the end of the sentence. Maybe give a range for the three clusters (except the PRD).
Pg 6 Line 15: What does this sentence mean "if only continuous records since 2013 are used in the analysis"? There is nothing in Section 2.1 about missing data or data prior to 2013. Figure 2 caption says "includes sites with partial records" but this detail should be in the main text too, not just in a caption. Pg 7 Line 17 mentions "including sites with partial records".
Pg 6 Line 20-29: Are any of these numbers from the tables either inset in Figure 2 or from Table 1? No table or figure is referenced except Figure 4 for the temperature trends. At least the 3.3 ppb a-1 is from Table 1/Figure 2 but if these numbers are otherwise new, would another table be helpful?

Pg 6 Line 29: move "2013-2019" out of "large increase" and move to "in temperature for 2013-2019 in June"

Pg 7 Line 5: add commas after conditions and the (Chen and Lu, 2016) reference

Pg 7 Line 9: how many days or percentage of days during the period are foehn vs no-foehn conditions? Are the composites for one heavily weighted over the other? It is hard to interpret what an 85% increase is without knowing the base number of days in June that were foehn conditions.

Pg 22 Line 8-9: Could include the contour line colors (and style for the dashed line for surface NOx) for panel b in the figure caption